

**A comparison of pre-Millennium eruption (946 AD) and modern temperatures**
**from tree rings in the Changbai Mountain, northeast Asia**
**Running Title: Millennial changes in temperature of Changbai Mt.**
Haibo Du[1], Michael C. Stambaugh[2], J. Julio Camarero[3], Mai-He Li[1,4], Dapao Yu[5],
Shengwei Zong[1], Hong S. He[2*], Zhengfang Wu[1*]
[1] Key Laboratory of Geographical Processes and Ecological Security in Changbai
Mountains, Ministry of Education, School of Geographical Sciences, Northeast
Normal University, Changchun 130024, China
[2] School of Natural Resources, University of Missouri, Columbia, Missouri, USA
[3] Instituto Pirenaico de Ecología, IPE-CSIC, 50059 Zaragoza, Spain
[4] Swiss Federal Institute for Forest, Snow and Landscape Research WSL, 8903
Birmensdorf, Switzerland
[5] CAS Key Laboratory of Forest Ecology and Management, Institute of Applied
Ecology, Chinese Academy of Sciences, Shenyang, 110016, China
*Correspondence to*: wuzf@nenu.edu.cn; HeH@missouri.edu



**Abstract:** High-resolution temperature reconstructions in the prior millennium are
limited in northeast Asia, but important for assessing regional climate dynamics. Here,
we present, for the first time, a reconstruction of April temperature for ~300 years
before the Millennium volcanic eruption in 946 AD, using tree rings of carbonized
logs buried in the tephra in Changbai Mountain, northeast Asia. The reconstructed
temperature changes were consistent with previous reconstructions in China and
Northern Hemisphere. The influences of large-scale oscillations (e.g., El
Niño-Southern Oscillation) on temperature variability were not significantly different
between the period preceding the eruption and that of the last ~170 years. However,
compared to the paleotemperature of the prior millennium, the temperature changes
were more complex with stronger temperature fluctuations, more frequent
temperature abruptions, and a weaker periodicity of temperature variance during the
last one and half centuries. These recent changes correspond to long-term
anthropogenic influences on regional climate.

**Keywords:** Carbonized logs; Changbai Mountain; dendroclimatology; Millennium
volcanic eruption; temperature reconstruction; tree rings.



**1. Introduction**

The observed global mean surface temperature for the decade 2011-2020 was ~1.09 °C higher than the average over the 1850-1900 period, reflecting the warming trend since pre-industrial times (IPCC, 2021). Century-wide predictions have been made based on relatively short-term observations, which bear great uncertainties especially at local and regional spatial scales. Different from the short instrumental records, reconstructing long-term climate variability using annually-resolved proxies such as tree rings is important for analyzing the long-term variations in climate and discriminating among natural and anthropogenic factors that drive climate change (Wang et al., 2018). Long-term, historical dendroclimatic reconstructions of temperature are essential to validate global climate models and provide important inputs to understand vegetation succession and vegetation-climate relationships in the region (Schneider et al., 2015).

Tree rings are excellent proxies for high-resolution climate reconstruction. Several millennial-scale annual climate reconstructions have been developed by multiproxy data for the globe(Consortium et al., 2013; Mann et al., 2008; Mann and Jones, 2003), North Hemisphere (Guillet et al., 2017; Moberg et al., 2005; Mann et al., 1999) or North Hemisphere extratropical regions (Schneider et al., 2015; Ljungqvist, 2010). Some millennial temperature reconstructions with very coarse temporal resolution using other proxies were also completed in northeast Asia, e.g., using varved sediment

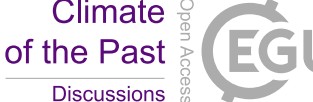

in Lake Sihailongwan (Chu et al., 2011). However, the millennial-scale and
high-resolution climate reconstructions rarely include tree-ring proxies from northeast
Asia due to limited available tree records prior to the last millennium.

The Changbai Mountain is the highest mountain in northeast Asia and encompasses
all life zones found along altitudinal gradients from temperate forests to the alpine
tundra (Zhou et al., 2005). Tree radial growth is sensitive to climate change in the
Changbai Mt which has allowed building several dendroclimatic reconstructions for
the past centuries (Lyu et al., 2016; Zhu et al., 2009; Shao and Wu, 1997). The highest
peak of Changbai Mt. with origins from an intraplate stratovolcano (Tianchi volcano)
located on the border between China and North Korea (Sun et al., 2014). A Plinian
eruption occurred around 1000 AD (well-known as the 'Millennium Eruption') with a
volcanic explosivity index of 7 based on an estimated eruptive column of ~25-35 km
and a total tephra volume of ~100 km$^3$ (Wei et al., 2003; Horn and Schmincke, 2000).
The eruption destroyed most plants within a ~50-km-radial area, but many trees
buried by volcanic ash became carbonized logs (Cui et al., 1997). These carbonized
logs provide a unique material to reconstruct climate of Changbai Mt. prior to the
millennium eruption using tree-ring records as climate proxies.

Numerous studies have attempted dating of the Millennium Eruption (Chen et al.,
2016; Xu et al., 2013; Yin et al., 2012). Recently, this eruption has been dated to the



end of 946 AD using a conspicuous dating marker of the ephemeral burst of
cosmogenic radiation in 775 AD (Oppenheimer et al., 2017; Büntgen et al., 2014) and
historical documents (Yun, 2013). With this date, carbonized trees provide the
opportunity to reconstruct climate before 946 AD in Changbai Mt., a region where the
greatest increase in air temperature over China was recorded during the last century
(Ding et al., 2007).

Here, we analyze tree rings from the carbonized logs and modern trees on Changbai
Mt. to reconstruct and compare temperatures between the three centuries
pre-Millennium Eruption and the last two centuries (1885-2012). These temperature
reconstructions can reveal the long-term regional climate dynamics in northeastern
China or even Northeast Asia and allow characterizing recent features related to
anthropogenic climate warming.

**2. Material and methods**
The Changbai Mt. ranges from 713 to 2691 m a.s.l., and belongs to the temperate
continental and mountain climate, with an annual mean temperature ranging from -7.3
to 4.9 °C and annual precipitation from 800 to 1800 mm (Du et al., 2018). The period
of cambial growth of trees is approximately May to September at low altitudes (e.g.,
1000 m a.s.l.) and shortens to June to August at high altitudes (e.g., treeline located at
~2060 m a.s.l.) (Du et al., 2021).



We sampled 55 carbonized trees from two nearby sites on the western slope of
Changbai Mt. in 2012 and 2013 (site A, 42°9′ N, 127°52′ E, 1025 m a.s.l., with 33
samples; and site B, 42°5.7′ N, 127°42.4′ E, 892 m a.s.l., with 22 samples) (Figure 1a,
b, c). Most of these trees had bark indicating the last year of tree growth was present
(Figure 1d). Radiocarbon dating of the wood of the outermost rings of two trees (from
sites A and B, respectively; Figure 1a) was conducted in the Accelerator Mass
Spectrometry (AMS) Laboratory at Peking University (Table 1) and indicated that
these trees died during the Millennium Eruption in 946 AD. Other carbonized trees
found and reported in previous studies were also dead in 946 AD (e.g., Oppenheimer
et al., 2017; Xu et al., 2013; Yin et al., 2012). Many of these carbonized tree samples
were not totally carbonized (Figure 1b) and showed a complete tree trunk (Figure 1c),
indicating that little or no transport has occurred from their original location.

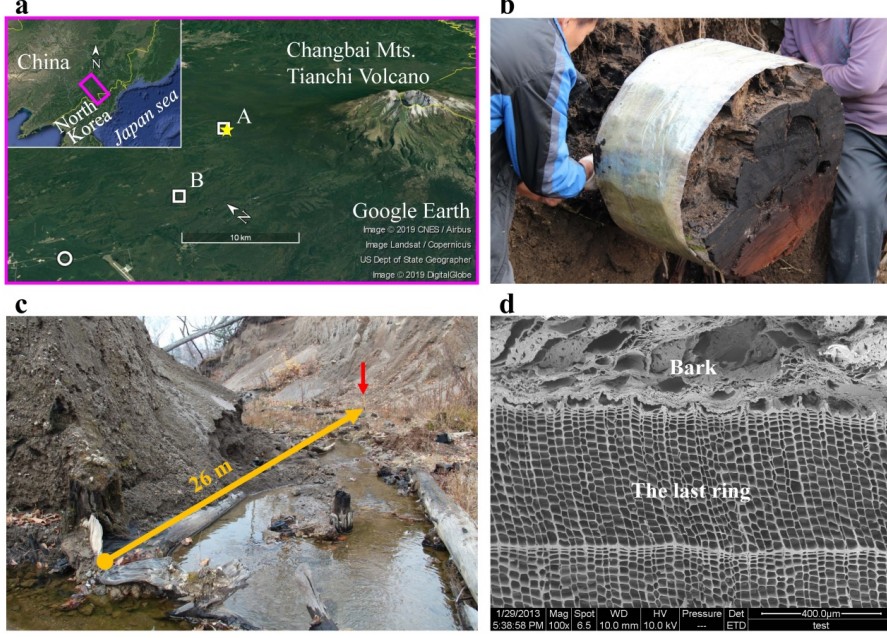


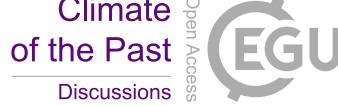

**Figure 1.** (**a**) Location of the Changbai Mountain and sample sites on Changbai Mt.
(from Google Earth Image). White squares represent sites where the carbonized logs
were found (A, Weidongzhan; B, Xiaoshahe). Yellow star shows the sampling site of
the modern forest. White circle indicates the Donggang National Datum
Meteorological Station. (**b** and **c**) Context of carbonized logs in the field. Species of
logs are *Pinus koraiensis*. (**d**) Cellular characteristics of the outermost tree ring and
bark of the carbonized log shown in (b).

**Table 1**. AMS $^{14}$C results of the complete outermost rings of two carbonized logs
collected from Weidongzhan (Site A) and Xiaoshahe (Site B) on the western slope of
the Changbai Mountain.

| Lab ID | Site | AMS $^{14}$C age (yr BP)* | Tree-ring calibration age (AD, 1σ (68.2%)) | Tree-ring calibration age (AD, 2σ (95.4%)) |
|--------|------|---------|----------------|----------------|
| BA150220 | A | 1155±20 | 780-790 AD (1.0%) | 770-970 AD (95.4%) |
| | | | 820-840 AD (7.4%) | |
| | | | 860-900 AD (33.8%) | |
| | | | 910-950 AD (25.9%) | |
| BA121692 | B | 1090±20 | 895-920 AD (24.8%) | 890-1020 AD (95.4%) |
| | | | 945-990 AD (43.4%) | |

* AMS $^{14}$C ages are dated at the Peking University AMS Laboratory and given in year
BP (years before 1950).




We identified the tree species of carbonized trees by analyzing microscopic
anatomical features of wood on three planes (cross-sectional, radial, and tangential)
(Figure S1). Eighteen of the 55 sample trees were Korean pine (*Pinus koraiensis*
Siebold & Zucc.). We used these trees to reconstruct the climate before the
Millennium Eruption (946 AD) using the current climate response of Korean pine
growth (Zhu et al., 2009). Prior to performing the climate response analyses, we also
sampled modern living Korean pines. Core samples from 27 living Korean pine trees
located near site A (see Figure 1a) were collected in 2013 and at 1.3 m height using a
Pressler increment borer.

Tree-ring width measurements and chronology development of carbonized and living
samples were conducted using standard dendrochronological techniques (Cook and
Kairiukstis, 2013). All available wood cross-sections/cores were first visually
cross-dated after cutting or sanding the wood surface, and then the quality of the
cross-dating was checked using the COFECHA program (Holmes, 1983). Age
detrending of the ring-width series was performed by fitting negative exponential
curves and calculating ring-width indices through the ratio method (Fritts, 1976).
Then, autoregressive modeling was used to remove first-order autocorrelation and to
obtain pre-whitened or residual series of ring-width indices. Standardized (STD) and
residual (RES) growth chronologies, with or without first-order autocorrelation



respectively, were developed by calculating robust biweight means using the
ARSTAN program version 49 (Cook et al., 2017; Cook, 1985). Then, subsample
signal strength (SSS) was used to evaluate the suitability and reliability of chronology
data for climate reconstructions (Buras, 2017; Wigley et al., 1984). The SSS > 0.85
was used to determine the robust and maximum chronology length and to ensure the
reliability of the reconstructions (Figure S2). This threshold corresponded to a
minimum sample depth of 12 samples for the carbonized tree chronology (from 746
AD) and 14 samples for the living tree chronology (from 1885 AD onwards).

Instrumental climate data were obtained for the period 1961-2012 from the Donggang
National Datum Meteorological Station (42°6′ N, 127°34.2′ E, 851 m a.s.l., situated
15-22 km apart from the sample sites; source: China Meteorological Data Network,
http://data.cma.gov.cn/). We calculated the Pearson correlation coefficients between
both chronologies and different time-scale (monthly, seasonal and annual) climate
variables (precipitation, mean temperature) to identify the main climate factors
driving tree growth. Correlations were calculated from prior April to current
September. Besides, to remove the trend effects, the correlation coefficients between
the first-order difference series of chronology and climate variables were also
calculated to further explore their relationships. Because the RES chronology (Figure
2) showed to be more highly correlated with climate than the STD chronology
(Figures S3-S5), we used the RES chronology to reconstruct temperature.



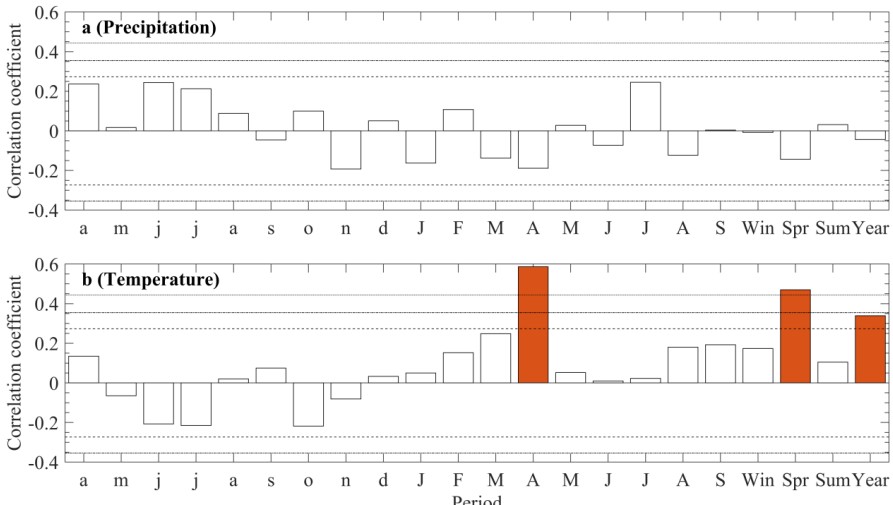

**Figure 2.** Pearson correlation coefficients between the RES tree-ring chronology and monthly, seasonal and annual (**a**) precipitation and (**b**) temperature during 1961-2012. Lowercase and uppercase letter on x axis indicate the months of the previous and current year, respectively. The horizontal dotted, dash-dotted, and dashed lines represent significance levels of 0.001, 0.01, and 0.05, respectively. Bars with significant correlation are filled with red colour.

We used a linear regression model to reconstruct past climate from the RES tree-ring chronology. The reliability of the regression model was evaluated using split sample calibration-validation statistics whereby calibration was conducted for 1961-1986 and validation was done for 1987-2012, after which the periods were switched and the process repeated (Table 2). Model statistics included the Pearson correlation coefficient, coefficient of determination ($R^2$), reduction of error (RE), coefficient of efficiency (CE), Durbin-Watson text (DW), root-mean-square error (RMSE), and

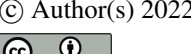



mean absolute error (MAE). Any positive RE/CE is generally accepted as indicative
of reasonable skill in the reconstructions (Cook et al., 1994; Briffa et al., 1988; Fritts,
1976). The DW statistic tests the temporal autocorrelation in the residuals between
modelled and observed climate data.

**Table 2**. Calibration/verification statistics of the temperature reconstruction.

|  | Calibration | Verification | Calibration | Verification | Calibration |
| --- | --- | --- | --- | --- | --- |
|  | 1961-1986 | 1987-2012 | 1987-2012 | 1961-1986 | 1961-2012 |
| Years | 26 | 26 | 26 | 26 | 52 |
| Correlation | 0.51 | 0.66 | 0.66 | 0.51 | 0.59 |
| $R^2$ | 0.26 | 0.43 | 0.43 | 0.26 | 0.34 |
| RE | -- | 0.40 | -- | 0.24 | -- |
| CE | -- | 0.35 | -- | 0.17 | -- |
| DW | 2.28 | 2.16 | 2.16 | 2.28 | 2.16 |
| RMSE | 1.43 | 1.44 | 1.35 | 1.52 | 1.41 |
| MAE | 1.13 | 1.13 | 1.01 | 1.19 | 1.10 |

Correlation, $R^2$, and DW were calculated between instrumental April temperature and
RES tree-ring width chronology. Reduction of error (RE), coefficient of efficiency
(CE), root-mean-square error (RMSE) and mean absolute error (MAE) were
calculated between instrumental and reconstructed April temperatures.





To analyze the abrupt changes in temperature between both periods, we calculated the
changes in mean state of temperature reconstructions using a heuristic segmentation
algorithm (the imposed minimum length of segments is 35 years) developed by
Bernaola-Galván et al. (2001). This method has been widely used to determine the
abrupt changes in mean state of a chronology (Gong et al., 2006). The significance of
the changes was estimated by the *t* test.

Power spectrum analysis was applied to investigate the reasonable periodicities in our
climate reconstructions. We used the wavelet analysis with a Morlet wavelet to
examine the periodicity of the reconstructed series and to check how periodicity
changes through time (Torrence and Compo, 1998). This analysis was separately
performed over the two ranges of the reconstructions. These analyses were carried out
using the Matlab R2019b software.

**3. Results and Discussion**
3.1. Temperature reconstruction
Precipitation in all months and seasons showed no significant correlation with the
RES chronology (Figure 2a). This was expected since the study area is wet, and
moisture-deficits that limit plant growth are uncommon. However, mean temperature
in April showed a positive and significant ($p < 0.001$) correlation ($r = 0.59$) with the
RES chronology during 1961-2012 (Figure 2b), indicating that the radial growth of





Korean pine is primarily limited by temperature in the month preceding cambial onset.
Moreover, the correlation coefficients for the first-order difference series indicated
that chronologies still have statistically significant and strongest relationship with
April temperature (Figures S4-S5). These are similar to the findings for Korean pine
growth found in north of Changbai Mt. (Zhu et al., 2009) and more broadly across
northeast Asia (Wang et al., 2017). Other species in other cold regions show
consistent growth responses to pre-growth temperature as a limiting factor in annual
radial growth (e.g., Hinoki cypress (*Chamaecyparis obtuse*) in central Japan
(Yonenobu and Eckstein, 2006), Georgei fir (*Abies georgei*) in the southeast Tibetan
Plateau (Liang et al., 2009), and Scots pine (*Pinus sylvestris L.*) in northern Poland
(Koprowski et al., 2012)). A positive growth response to April temperature can occur
due to warming in the period prior to the growing season causing tree dormancy to
break early, accelerating the division and enlargement of cambial cells, and extending
the length of the growing season (Schweingruber, 1996). The correlation between the
RES chronology and spring temperature is also significant, mainly due to the effects
of April temperature.

The positive RE (0.40) and CE (0.35) statistics for the late verification period and
positive RE (0.24) and CE (0.17) statistics for the early verification period indicated
reasonable reconstruction skill for both compared sub-periods (Table 2). Therefore,
we used the full calibration period for developing April temperature model by



calculating the model $R^2$, DW (Durbin-Watson statistic), RMSE, and MAE (Table 2).
The DW statistic calculated over the full calibration period ($\text{DW}_{1961\text{-}2012} = 2.16$, $p <$
0.01), achieving a value close to the optimal value of 2 which indicates no significant
autocorrelation in the residuals. The full tree-ring model predicting April temperature
was given as:
$$y = 7.38*RES - 1.79 \qquad\qquad\qquad (1);$$
where $y$ is April temperature and $RES$ is the residual tree-ring index, being the model
and the predictor variables significant at $p < 0.01$. This regression model accounted
for 34% of the variance in instrumental April temperature (Figure 3b). The
reconstructed April mean temperature showed decreasing trend after 2000, which was
different from the results of reconstructed April-July minimum temperature by Lyu et
al. (2016) and February-April temperature by Zhu et al. (2009). However, the
decreasing trend was coincident with the change in the observed April temperature
(Figure 3a). Therefore, the difference of the change in temperature during the last
decade between the temperature reconstructions may be due to seasonal diversity or
regional difference. Besides, although the reconstruction underestimates the extreme
temperatures recorded in some years (e.g., 1965 and 1998), it successfully captures
both high and low frequency variations of temperature variability (Figure 3a).





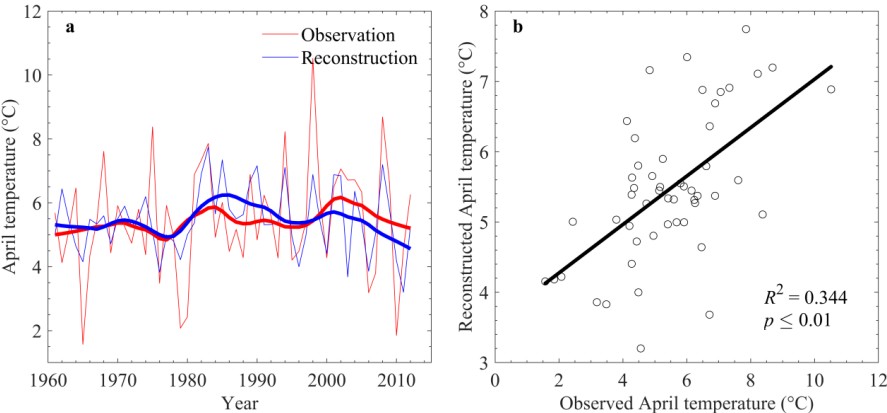


**Figure 3.** (**a**) Observed (red thin line) and reconstructed (blue thin line) annual April
temperature. Heavy lines are the corresponding 13-year moving averaged
temperatures. (**b**) Linear regression between observed and reconstructed temperatures
during the period 1961-2012.

3.2. Comparisons between changes in paleoclimate and modern climate
Based on the regression model, we reconstructed the annual April temperature and a
13-year moving average for periods 652-946 AD and 1830-2012 AD (Figure 4a).
Truncated periods of reconstructions where SSS is > 0.85 were 746-946 AD and
1885-2012 AD. Although our temperature reconstructions did not match well with the
previous temperature reconstructions using the varved sediment in Lake Sihailongwan
in the Changbai Mt. (Chu et al., 2011), both temperature reconstructions showed
coincident decreasing-increasing-decreasing variation during 850-946 AD. The lack
of agreement between these two proxies may be due to age model error inherent in
radiocarbon-dated records (Conroy et al., 2010). For regional scale, our temperature





reconstructions generally coincided with the variations of pre-millennial temperature
in China (Yang et al., 2002). Interestingly, variations in April temperature
reconstructions in both two periods in this study are similar to those observed in
summer temperature reconstructions for Northern Hemisphere (Guillet et al., 2017).
These temperature reconstructions all display warm periods in 830-850 AD, 885-895
AD, 1931-1953 AD, 1981-2000s AD, and cold periods in 1830-1840 AD and
1955-1980 AD.

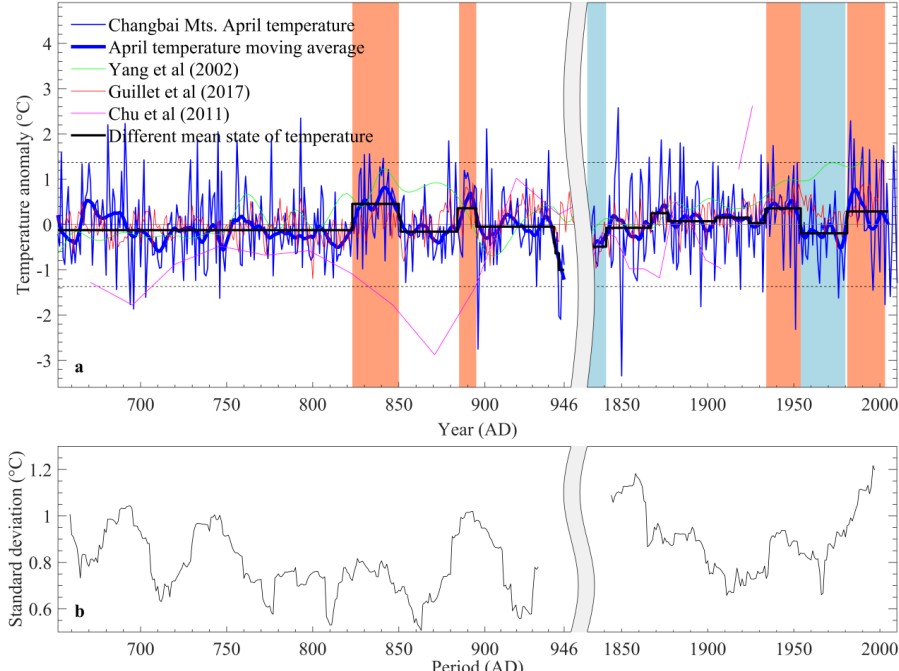


**Figure 4.** (**a**) Reconstruction of April temperature anomaly (blue thin lines) for
652-946 AD and 1830-2012 AD, with a 13-year moving average (blue bold lines) for
Changbai Mt. The temperature anomaly is relative to the mean during the entire
period. Green, red, and magenta lines indicate standardized temperature anomalies in



China ("H-res") (Yang et al., 2002), the summer temperature anomalies for Northern
Hemisphere (Guillet et al., 2017) and temperature anomalies using the varved
sediment in Lake Sihailongwan in the Changbai Mt. (Chu et al., 2011), respectively.
Black bold lines show the changes in mean state of the April temperature
reconstructions. (**b**) Time series of standard deviation of 30-year moving window for
the April temperature reconstructions in this study. A given period (e.g., 900 AD)
represents a standard deviation in 30 years before and after that period (e.g., 886-915
AD).

During 746-946 AD, the standard deviation (s.d.) of reconstructed mean April
temperature was 0.77 °C, whereas the standard deviation was 0.92 °C for the period
1890-2012. That is, temperature variance (standard deviation) increased 18% for
modern temperature in comparison to the temperature prior to the 946 AD Millennium
Eruption. Moreover, the 30-year moving standard deviations showed periodical
change and smaller values during the pre-eruption period than those in the last 180
years (Figure 4b). Specifically, the 30-year moving temperature variance showed
significant ($p < 0.05$) periodicity of 50-70 years (Figure S6ab). However, there was no
significant periodicity of temperature variance during the last 180 years (Figure
S6cd).

There were only five periods with significant differences in mean state of temperature





during the last 200 years before the volcano eruption (Figure 4a). The two warm
periods in 830-850 AD and 885-895 AD were widely recognized as warm epochs also
by other temperature reconstructions for Northern Hemisphere extratropical areas
(e.g., Esper et al., 2002). In contrast, nine periods with significantly different mean
temperature states were revealed during the last ~180 years before present. Moreover,
only nine warm years (defined as > 1.5 s.d.; years: 756, 776, 793, 830, 833, 841, 879,
901, 937) and 5 cold years (defined as < 1.5 s.d.; years: 746, 896, 930, 943, 944) were
identified during the last 200 years before the Millennium Eruption, whereas 19 warm
years (1847, 1848, 1866, 1873, 1878, 1884, 1886, 1903, 1931, 1938, 1982, 1983,
1985, 1990, 1994, 1998, 2001, 2002, 2008) and 10 cold years (1850, 1896, 1919,
1932, 1951, 1976, 1996, 2003, 2006, 2011) were identified during the last 170 years
before present (Figure 4a).

These differences may be partly due to the anthropogenic influences since
approximately the beginning of the Industrial Revolution (Gong et al., 2006). For
example, the probability of present-day hot extremes increased 1-1.2% relative to
pre-Industrial Revolution time in the region of Changbai Mt. Presently, 75% of the
moderate hot extremes occurring worldwide are attributable to climate warming, of
which the majority are extremely likely to be anthropogenic (Fischer and Knutti,
2015). Higher 30-year standard deviations during the last 180 years than the
pre-eruption period may also support the attribution to increased anthropogenic





influences on thermal conditions (Figure 4b).

Wavelet analysis indicated significant ($p < 0.05$) periodicity of ~ 4.5 years during
770-946 AD (Figure 5ab) and ~ 3.6 years during the last ~120 years (Figure 5cd).
Similar periodicities of 3 to 4 years were also found in previous analyses of
instrumental and reconstructed temperatures (Zhang et al., 2013; Yu et al., 2013; Chen
et al., 2010). Although our temperature reconstructions do not contain the significant
($p < 0.05$) quasi-11-year periodicity (e.g., Li et al., 2011) during the entire period, the
significant ($p < 0.05$) quasi-11-year periodicity in 880-910 AD and the non-significant
quasi-11-year period in 785-810 AD and 850-870 AD were found. Significantly short
periodicities are typically associated with El Niño-Southern Oscillation (ENSO)
(Stone et al., 1998; Allan et al., 1996) and temperature in March to May in northeast
China is affected by ENSO (Yuan and Yang, 2012). Moreover, the changes in
temperature in northeast China correspond to the quasi-4-year changes in sea surface
temperature in the central and eastern Pacific Ocean, suggesting a close link between
temperature variation in northeast China and the ENSO cycle (Zhu et al., 2004).
These results indicate that the effects of some large-scale oscillations (e.g., ENSO) on
paleo- and modern- temperature continue to be important to the climate forcing in the
region of Changbai Mt.



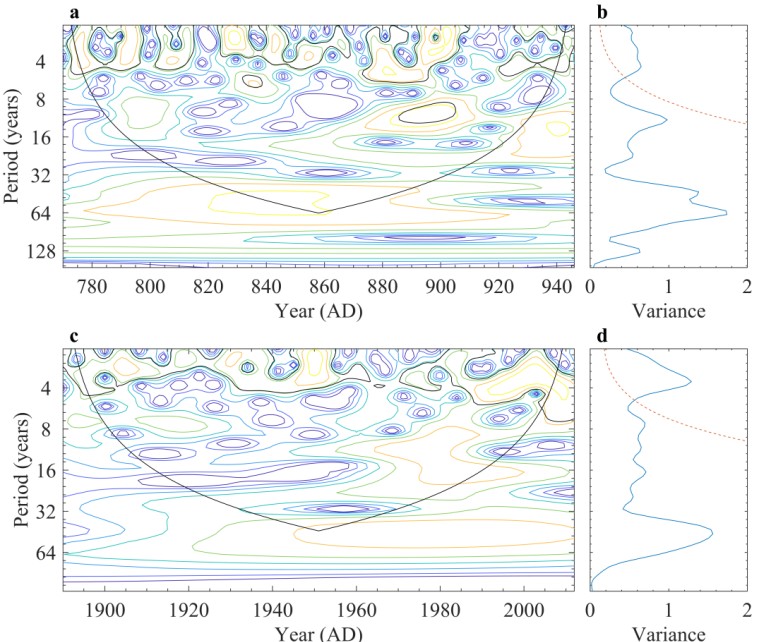

**Figure 5.** (**a**, **c**) Wavelet power spectrum of the reconstructed April temperature from

(a) carbonized and (c) modern trees. The power has been scaled by the global wavelet

spectrum. Black contour is the 95% confidence level using a red-noise (autoregressive

lag1) background spectrum. (**b**, **d**) The global wavelet power spectrum (light blue line)

for (b) carbonized trees-based and (d) modern trees-based temperature reconstruction.

Dashed lines represent a significance of 0.05.

**4. Conclusions**

We presented a new 295-year reconstruction of April temperature before the

Millennium Eruption occurred in 946 AD at Changbai Mt. using unique tree-ring

proxies of carbonized logs buried in the tephra, which was compared to that of living

trees growing during the last 183 years. Temperature reconstructions correspond well



with previous large-regional temperature reconstructions. Our results showed that,
although the influences of some internal variability (e.g., ENSO) on variation in
temperature do not change between the periods, the changes in modern temperatures
become more complex (e.g., increased variation and abrupt changes, and weakening
in periodicity of temperature variance) than those in period prior to 946 AD likely due
to anthropogenic influences. The present study provides tree-ring proxies for climate
reconstructions in northeast Asia for the last millennium. Documentation of these
features is important for understanding long-term regional climate dynamics and
analyzing the millennial-scale changes in vegetation-climate relationship in northeast
Asia.



**Data availability**
Tree-ring chronology and climate data used in this study are archived at ZENODO:
https://doi.org/10.5281/zenodo.6633856. They can also be provided by the
corresponding author.

**Authors' contributions**
H.D., Z.W., and H.S.H. conceived and led this study. Tree rings and observation data
collection was led by H.D. and S.Z. Analyses and the first draft was carried out by
H.D. and M.S., and all authors contributed to revising subsequent manuscripts.

**Declaration of Competing Interest**
The authors declare that they have no conflict of interest.

**Acknowledgments**
We acknowledge support from the Joint Fund of National Natural Science Foundation
of China (U19A2023) and the Fundamental Research Funds for the Central
Universities (2412020FZ002).



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
