# Peer review of "A comparison of pre-Millennium eruption (946 AD) and modern temperatures"

_Climate of the Past, 2022_

## Author Comment (AC1)

Reviewer#1:

*The authors developed a floating tree-ring width chronology covering ca. 300 years in Middle Age and ca.200 years in modern time, using living trees and carbonized logs in Changbai Mountain, northeast Asia and then reconstructed air temperatures based on these tree-ring materials in this region. The work with the materials of tree rings from carbonized logs is important to the studies of climate, but the shortcomings of this manuscript are also clear, especially in quality control and the standardization of the data. Here, some main points need to be addressed:*

**[Response]**: We are very grateful to you for the approval of the significance of this work. Most importantly, we would like to thank you for pointing out the shortcomings including the lack of the information on the quality statistics of the chronologies, missing reliable information for the species identification, reconstructing temperature using the RES chronology, and some confused statements when referring the previous study. According to your comments and suggestions, we have added all the relevant information and clarified the relationship between our study and the study by Zhu et al., (2009).

Again, thank you very much for these valuable and insightful comments and suggestions, which helped us to improve the MS quality significantly. Below you will find our point-to-point response.

**1.** *Dendroclimatology is based on the high quality cross-dating tree-ring data and reliable fidelity of growth-climate response. The cross-dating, especially the process and results, is the most crucial step in the dendrochronological study. However, the authors seemed deliberately evading the question, and did not give sufficient details on their samples, for example:*
*1) line128-129, authors did not show the clear information that "these trees" were 18 Korean pines or the total of 55 trees.*
*2) line 148-157, authors presented that they did cross-dating, tree-ring width measurements and chronology development, however, they did not illustrate the quality control criteria of the cross-dating, measuring process, measurement accuracy and instruments, quality statistics of the chronologies (e.g., the correlation between cores and correlation between trees and so on) and the main statistics of chronologies.*
*Without the information on the quality statistics, the readers and the referees, cannot give an objective evaluation for your manuscript, and the subsequent results, analyses, discussions, and conclusions in this manuscript become doubtful.*

**[Response]**: We apologize to the confusing information of carbonized trees used to reconstruct temperature in the previous version. Korean pine was the most species (18 trees) identified from the 55 carbonized trees. To ensure the quality of temperature reconstruction, we decided to use the same species, Korean pine, to build the chronologies for both the pre-Millennium and modern periods. However, some small Korean pine was excluded from the chronology development, and only 19 cores from ten Korean pines after the quality control of the cross-dating were finally used for developing chronology. The exact information is that: "*We used 19 cores from ten carbonized Korean pine trees to reconstruct the climate before the Millennium Eruption (946 AD).*" We also supplemented the information of these cores/trees in Table 2 in the revised manuscript.

We also added the information on the quality control of the cross-dating, measuring process, measurement accuracy and instruments, and quality statistics of the chronologies in the revised manuscript. The updated information is showed below (Lines 142-165): "*Tree-ring width measurements and chronology development of carbonized and living samples were conducted using standard dendrochronological techniques (Cook and Kairiukstis, 2013). The carbonized and living samples were naturally air-dried. The surface of the stem cross section of carbonized trees and the core of living trees was polished with a sandpaper polishing machine, and thick and thin brushes. Then, the tree ring width was identified and recorded by a LINTAB 6 measuring system with an accuracy of 0.001 mm. For each carbonized tree, two cores along one line crossing the pith were measured. Quality control of measurements and cross-dating was assessed using the COFECHA software (Holmes, 1983). Core segments having low correlation with the master chronology were excluded from the analysis. Tree ring width series were detrended using polynomial functions (splines with a period of 67% of series length). However, results may be sensitive to the detrending method (Peters et al. 2015). Therefore, to ensure robustness of our results to method choice, age detrending of the ring-width series was also performed by fitting negative exponential curves. Standardized (STD) growth chronologies for carbonized and living samples were developed by calculating robust biweight means using the ARSTAN program version 49 (Cook et al., 2017; Cook, 1985). Then, subsample signal strength (SSS) was used to evaluate the suitability and reliability of chronology data for climate reconstructions (Buras, 2017; Wigley et al., 1984). The SSS > 0.85 was used to determine the robust and maximum chronology length and to ensure the reliability of the reconstructions (Figure S2). This threshold corresponded to a minimum sample depth of 11 samples for the carbonized tree chronology (from 745 AD) and 13 samples for the living tree chronology (from 1883 AD onwards) (Table 2). The dendrochronological characteristics of the STD ring-width chronologies of carbonized and living trees were showed in Table 2.*"

**For tree species identification of carbonized logs:**
*2. Line126-128, the authors did not give a reliable information for the species identification. Authors should present the main anatomical features of their samples matching Korean pine wood attributes briefly and professionally in the manuscript because some readers may not have sufficient wood anatomy knowledge. The presentation in sentence like line126-128, and the three planes (cross-sectional, radial, and tangential) in Fig S1 are insufficient to show the species being Korean pine.*

**[Response]**: We thank you for this suggestion. In the revised manuscript, we presented the main anatomical features of the tree planes of carbonized samples, which were used to match Korean pine wood attributes. This information was showed in the section "**Identifying Korean pine tree species from carbonized trees**" in the Supplement file. We also showed the anatomical features of the three planes for all the 18 identified Korean pine trees (Figure S1) in the Supplement file.

We also enclosed the content of "**Identifying Korean pine tree species from carbonized trees**" below:

**Identifying Korean pine tree species from carbonized trees**

In this study, the tree species of carbonized trees was identified by analyzing the microscopic anatomical features of wood on three planes (cross-, radial-, and tangential-section). For the identified Korean pine (*Pinus koraiensis* Siebold & Zucc.) in this study, the microscopic anatomical features were showed below:

**Cross-section features**. The boundary of growth ring is slightly obvious, and the early wood gradually changes to late wood within the ring. The growth of early wood is uniform, and the early wood accounts for most of the ring width. The axial parenchyma tissue is absent. The resin channels are divided into axial and radial types. The axial channels are mostly single and usually distributed in the late wood or located at 2/3 of the ring width. The lipid cells are thin-walled and often contain quasi-invasion bodies. There are 3~6 lipid cells around the maximum axial resin channel. The transverse section of early wood tracheid is square, rectangular and polygon, and the axial parenchyma tissue is absent.

**Radial-section features**. Radiative tracheids are located in 1~2 columns of the upper and lower edges, and are occasionally seen in the middle of the rays. The xylem low rays are sometimes made up entirely of radial tracheids. The inner wall is smooth and the outer edge is wavy. The cross field pattern between ray parenchyma cells and early wood tracheids was pane-like, or pine type was occasionally seen in late wood,

with number of 1~4, 1 (rare 2) in horizontal line. Axial parenchyma cell horizontal wall perforation and end wall nodular thickening are usually absent or few, not obvious, no indentation.

**Tangential-section features**. There are 4-8 wood rays per mm, including two types: single column and spindle-shaped. The single column of rays is 2~13 cells or more in height, most of them are 3~8 cells; the spindle rays have radial resin canals, 2~3 columns of ray cells above and below the approach; The upper and lower ends gradually sharpen into a single column, the height of which is 3-10 cells or more. The ray cells are usually oval in shape. The radial resin tract is much smaller, with 3-5 adipocytes around the tract.

**Conclusions**. According to the microscopic anatomical characteristics of the above three planes, the carbonized logs, No. CBSXA-01, CBSXA-12, CBSXA-21, CBSXA-24, CBSXA-27, CBSXB-01, CBSXB-02, CBSXB-03, CBSXB-05, CBSXB-07, CBSXB-08, CBSXB-10, CBSXB-11, CBSXB-14, CBSXB-15, CBSXB-17, CBSXB-21, and CBSXB-26, were identified as Korean pine (*Pinus koraiensis*) of the family *Pinaceae* and the genus *Pinus*. The anatomical features of the tree planes of the 18 samples were separately showed in the large Figure S1.

**For the reconstruction:**
**3.** *RES chronology emphasizes high-frequency variations using autoregressive prewhitening that removes autocorrelation from the series, but temperature usually contains multi-band signals, especially low frequency information, with high autocorrelation. Thus, reconstruction of temperature variables using RES chronology with only the statistical correlation sounds unscientific, and lack of biophysical basis.*

**[Response]**: We sincerely thank you for this comment. And, we agree with you that RES chronology emphasizes high-frequency variations in temperatures. Therefore, we reconstructed temperature using STD chronology in the revised manuscript, and this revision did not affect the main findings of this study. Thank you again!

**4.** *In addition, authors used the documented growth-climate response with the current climate response of Korean pine by Zhu et al., (2009) as a current climate response of the pine in this manuscript. Do you and Zhu et al., (2009) use the same samples in the manuscript? If not, it is unscientific and unacceptable to use the assessed and verified data to match or as the using base of your unassessed new data.*

**[Response]**: We apologize for the confusing expression in the previous version. We used the growth-climate response of Korean pine by our samples **rather than** by Zhu et al. (2009) as the current climate response of the pine in our study. For example, we presented that "**Prior to performing the climate response analyses, we also sampled modern living Korean pines. Core samples from 27 living Korean pine trees located near site A (see Figure 1a) were collected in 2013**" (Lines 135-137). We wanted to express that Korean pine was previously successfully used to reconstruct temperature in Changbai Mountain (Zhu et al., 2009). Therefore, Korean pine is suitable for temperature reconstruction in this region. To avoid the previous confusion, we deleted the citation in this sentence, and moved this sentence to the position following the sentence which mentioned the samples of living Korean pines (Lines 138-140). Thank you for your comments!

---

## Author Comment (AC2)

Reviewer#2:

*The authors successfully reconstructed temperatures in Northeast Asia using buried carbonized logs and nearby living trees, and it looks very interesting, providing a data window into our understanding of the temperature before the millennium eruption of the Changbai Mountain volcano. But I still have some concerns. First, the description of the exact dating process of carbonized logs is not complete enough. Secondly, the method of establishing the width chronology needs to be improved. The RES chronology is not as good as the STD chronology in the recording of low-frequency signals. Finally, the teleconnection mechanism discussion simplicity requires more pattern process validation.*

**[Response]**: We thank you very much for the approval of the significance of this work. We also thank you for pointing out the three-aspect shortcomings and for the insightful and constructive comments and suggestions. We have carefully revised/corrected all the questions, which helped us to improve the MS quality significantly. Below you will find our point-to-point response.

**1.** *Lines 27~31 The time boundary is confusing, the specific time should be given in the past millennium and the past half century.*

**[Response]**: Thank you for your suggestion. We have added the specific time of both periods in the revised manuscript as: "…between the periods of 745-946 AD preceding the eruption and 1883-2012."

**2.** *Line 39 Why is it pre-industrial.*

**[Response]**: Thank you for your question. It should be "the industrial period". We have revised the statement in the revised manuscript.

**3.** *Line 45 Long-term and historical express the same meaning.*

**[Response]**: We deleted "historical" from the sentence. Thank you!

**4.** *Line 50 Add references.*

**[Response]**: We supplemented the classical book, "Tree rings and climate" (Fritts, 1976).

**5.** *Line 57 Gives the latitude and longitude position.*
**[Response]**: Done!

**6.** *Line 128 unified into English or Arabic numerals.*

**[Response]**: Thank you for your comment! We spelled small numbers (e.g., numbers less or equal to 10) out, but showed other as Arabic numerals. However, it is little proper to start a sentence with an Arabic numeral. Therefore, we spelled out the starting number "18" (Eighteen) but not spelled out the number "55" within the sentence.

**7.** *Lines 129~131 This sentence should specifically describe or list more references to support this method.*

**[Response]**: This sentence was a bit confusing, which was also pointed out by Reviewer#1. We wanted to express that: "Korean pine was previously successfully used to reconstruct temperature in Changbai Mountain (Zhu et al., 2009). Therefore, Korean pine is suitable to reconstruct temperature in this region. We used the growth-climate response of Korean pine by our samples rather than by Zhu et al. (2009) as the current climate response of the pine in our study."

To avoid this confusion, we deleted the citation in this sentence. We changed the relevant sentences as: "*Prior to performing the climate response analyses, we also sampled modern living Korean pines. Core samples from 27 living Korean pine trees located near site A (see Figure 1a) were collected in 2013 and at 1.3 m height using a Pressler increment borer. We used the 19 carbonized Korean pine cores to reconstruct the climate before the Millennium Eruption (946 AD) using the current climate response of Korean pine growth.*" (Lines 135-140) Thank you for your comments!

**8.** *Lines 158-161 What I want to know is the response of the chronology to the mean minimum temperature and mean maximum temperature, but this does not.*

**[Response]**: Considering your suggestion, we added the analysis of the response of the STD chronology to maximum temperature and minimum temperature in Figure 2. The STD chronology was still most sensitive to April mean temperature among the correlation relationships between the chronology and precipitation, mean temperature, maximum temperature, and minimum temperature.

[Figure]

Figure 2.

**9.** *Lines 161-162 Why is it from April of the previous year to September of the current year? Is there any relevant explanation?*

**[Response]**: Climate may show time-lag effects on tree radial growth (Zhou et al., 2022). Moreover, Korean pine is limited by temperature from the month preceding cambial onset onward (Wang et al., 2017; Zhu et al., 2009) and until the end of growing season (September). Therefore, we analyzed the correlations of tree radial growth and climate during the previous April to current September. We have added this information in the revised manuscript (Lines 176-179).

**10.** *Lines 164-166 Why does the RES chronology respond better? Is it caused by the different detrending methods used? I want to know about the chronology established by other detrending methods such as spline function or smooth curve. The RES chronology usually suffers from the loss of low-frequency signals, especially for temperature reconstruction, which cannot capture interdecadal signals.*

**[Response]**: We sincerely thank you for this comment. Maybe because RES chronology emphasizes high-frequency variations in temperatures, which caused higher correlation with short-term temperature (e.g., ~50 years) compared with STD chronology. However, we agree with you that RES chronology usually suffers from the loss of low-frequency signals, especially for temperature reconstruction. Therefore, to capture interdecadal signals, we reconstructed temperature using STD chronology in the revised manuscript, and this revision did not affect the previous main findings.

In the revised manuscript, we used polynomial functions (splines with a period of 67% of series length) to detrend the tree-ring width series. However, results may be sensitive to detrending method (your comments and Peters et al. 2015). Therefore, to ensure robustness of our results to method choice, we also used negative exponential functions to remove non-climatic signals in ring width. The results based on the negative exponential curve (Figure S5) are very similar to the results shown in the main text (Figure 2), which confirms the robustness of our findings. Thank you again!

[Figure]

Figure S5. Same as Figure 2, but for the detrending method of negative exponential curves.

**11.** *Lines 168-173 Has the author considered using a sliding correlation to test the stability of the temperature or precipitation signal?*

**[Response]**: Thank you for your comment. We added an analysis of the sliding correlation in the revised version. The results showed that the 30-year moving correlation coefficient between the STD chronology and April mean temperature did not change ($p > 0.01$) during 1961-2012 in the Changbai Mt (Figure S4).

[Figure]

Figure S4. Thirty-year moving correlation coefficient between the STD chronology and April mean temperature during 1961-2012 in the Changbai Mt. The thick dash line shows the average of the moving correlation coefficient.

**12.** *Lines 248-250 Has the author done a spatial comparative analysis? e.g., spatial correlation.*

**[Response]**: We did not calculate the spatial correlation between our temperature reconstructions with other reconstructions. However, we analyzed the correlation and regression relationships between our reconstructed temperature and the observed temperature in the Changbai Mt. (Figure 3).

**13.** *Lines 259-275 I suggest that the author add the millennium simulation results in the Earth coupling model (CESM) to increase the accuracy, stability and regional representativeness of the reconstruction results on the basis of the proxy data reconstruction comparison.*

**[Response]**: We sincerely thank you for this suggestion! We will use the Community Earth System Model 2 to simulate the entire millennial temperature time series based on the two-period reconstructed temperatures. We think that this should be a new work and will further improve our understanding of the changes in regional climate in Northeast Asia at millennial scale. In this study, however, the previously published temperature reconstructions with different space-time scales from the Changbai Mt., China, and Northern Hemisphere were adequate for the comparison with our

temperature reconstructions. Therefore, we want to use your suggestion for another further study based on this study. Thank you very much for this valuable and constructive suggestion!

**14.** *Lines 315-323 There are many ways to detect attribution, why didn't the author use it, such as multiple regression to calculate contribution, etc., but rely on simple statistics of increase and decrease.*

**[Response]**: We agree with you that the multiple regression model is a very useful approach to quantify the (relative) contributions of independent variables on the dependent variable in the model, e.g. by the slope and the $R^2$ of each normalized independent variable. In this study, however, we cannot get the anthropogenic factors prior to the 946 AD Millennium Eruption, and separately analyzing the regression relationship between temperature and anthropogenic factors exceeds the purpose of this study. Therefore, we used the simple but useful statistics to discuss the differences between the two periods.

**15.** *Lines 325-329 Maybe cross wavelet power spectrum XWT analysis is more suitable for the similarities and differences in the periodic changes of the two-time series.*

**[Response]**: Thank you for this suggestion. The XWT analysis is indeed useful to reveal the correlation and consistency of two different time scales and can reproduce the phase relationship in the time-frequency space. However, this method requires that the two time series must overlap. For example, at least four data available in the common time period are required in the Matlab codes provided by Grinsted (2023). In this study, the two temperature time series have no common time period. Therefore, we used separately the Wavelet analysis for the temperature reconstructions of the two different periods.

However, we supplemented an analysis of the correlation and consistency of the periodicity of the temperature in Changbai Mt. and SST in Eastern Tropical Pacific **based on the XWT**. This is also our response to your next comment below. Thank you for the suggestion again!

**16.** *Lines 335-341 The discussion on the SST model is too simple. Has the author verified such a teleconnection mechanism with the reanalysis data (NOAA, ERA)?*

**[Response]**: Thank you for this comment. We added discussion on the correlation and consistency of the periodicity of the temperature in Changbai Mt. and SST in Eastern

Tropical Pacific based on the XWT to verify the teleconnection mechanism in our study (Lines 363-367; Figure S7). The SST data are from NOAA Extended Reconstructed Sea Surface Temperature (ERSST) Version 5.

[Figure]

Figure S7. Cross wavelet transform of the spring mean air temperature of Changbai Mountain during 1961-2012 and mean sea surface temperature (SST) of Eastern Tropical Pacific (0 to 10 °South and 90 °West to 80 °West) time series (https://psl.noaa.gov/data/timeseries/monthly/NINO12/). The 5% significance level against red noise is shown as a thick contour. The relative phase relationship is shown as arrows (Grinsted et al., 2004; https://doi.org/10.5194/npg-11-561-2004).

**17.** *Figure 4. Needs to separate independent reconstructions and other comparison sequences, it looks very messy and cannot be discerned.*

**[Response]**: Thank you for this suggestion. We have revised Figure 4 according to your suggestion in the revised manuscript.

[Figure]

Figure 4.

**18.** *Figure 5. Contour lines should be displayed in color to identify significant periods more intuitively.*

[Response]: Thank you! We used bold red contour lines to identify significant periods in the revised Figure 5 and Figure S6.

[Figure]

Figure 5.

**References:**

Grinsted, A.: Cross wavelet and wavelet coherence (https://github.com/grinsted/wavelet-coherence), GitHub. 2023

Grinsted, A., Moore, J. C., and Jevrejeva, S.: Application of the cross wavelet transform and wavelet coherence to geophysical time series, Nonlin. Processes Geophys., 11, 561-566, https://doi.org/10.5194/npg-11-561-2004, 2004.

Peters, R. L., Groenendijk, P., Vlam, M., and Zuidema, P. A.: Detecting long-term growth trends using tree rings: a critical evaluation of methods. Global Change Biol., 21, 2040-2054, https://doi.org/10.1111/gcb.12826, 2015.

Wang, X., Zhang, M., Ji, Y., Li, Z., Li, M., and Zhang, Y.: Temperature signals in tree-ring width and divergent growth of Korean pine response to recent climate warming in northeast Asia, Trees, 31, 415-427, https://doi.org/10.1007/s00468-015-1341-x, 2017.

Zhou, Y., Yi, Y., Liu, H., Song, J., Jia, W., and Zhang, S.: Altitudinal trends in climate change result in radial growth variation of Pinus yunnanensis at an arid-hot valley of southwest China, Dendrochronologia, 71, 125914, https://doi.org/https://doi.org/10.1016/j.dendro.2021.125914, 2022.

Zhu, H. F., Fang, X. Q., Shao, X. M., and Yin, Z. Y.: Tree ring-based February–April temperature reconstruction for Changbai Mountain in Northeast China and its implication for East Asian winter monsoon, Clim. Past, 5, 661-666, https://doi.org/10.5194/cp-5-661-2009, 2009.